# REALISM INDEX: INTERPOLATION IN GENERATIVE MODELS WITH ARBITRARY PRIOR

## ABSTRACT

In order to perform plausible interpolations in the latent space of a generative model, we need a measure that credibly reflects if a point in an interpolation is close to the data manifold being modelled, i.e. if it is convincing. In this paper, we introduce a *realism index* of a point, which can be constructed from an arbitrary prior density, or based on FID score approach in case a prior is not available. We propose a numerically efficient algorithm that directly maximises the realism index of an interpolation which, as we theoretically prove, leads to a search of a geodesic with respect to the corresponding Riemann structure. We show that we obtain better interpolations than the classical linear ones, in particular when either the prior density is not convex shaped, or when the soap bubble effect appears.

## 1 INTRODUCTION

Since the advent of the Variational Auto-Encoder (VAE) (Kingma & Welling, 2013) and the Generative Adversarial Network (GAN) (Goodfellow et al., 2014), generative models became an area of intensive research, with new models being developed (e.g. Kingma & Dhariwal (2018); Larsen et al. (2015); Tolstikhin et al. (2017)). In these models, the data distribution is mapped into the latent space. An important profit from introducing the latent space is the ability to construct interpolations, i.e. traversals between latent representations of two different objects. Meaningfulness of decoded interpolations is often used as a supporting argument for networks generalisation capability (Bowman et al., 2015; Dumoulin et al., 2016).

Interpolation is used commonly to show that models do not overfit, but generalise well (Dumoulin et al., 2016; Goodfellow et al., 2014; Higgins et al., 2016; Kingma & Welling, 2013). Intuitively, a good interpolation should decode to meaningful objects, give a gradual transformation, and reflect the internal structure of the dataset. More precisely, we require that the interpolation curve (after transporting to the input space) is smooth and relatively short, while at the same time it goes through regions of high probability in the latent space (see leftmost projection in Fig. 1). However, in some cases, even for a Gaussian prior, a linear interpolation could be of poor quality, e.g. due to the so called "soap bubble effect" (Husar, 2017; Lesniak et al., 2019). This may result in low quality samples in the middle of the path (White, 2016). The above argument puts the usability of simple linear interpolation in question and motivates further research in this area (Agustsson et al., 2017; Brock et al., 2016; Kilcher et al., 2017; Larsen et al., 2015; Lesniak et al., 2019; White, 2016).

In this paper, we construct a general interpolation scheme, which works well for arbitrary priors. We introduce a notion of a *realism index* of an element of the latent space, which naturally generalises to arbitrary curves. We show that in general the proposed method can be regarded as a search for geodesics in a respectively modified local Riemann structure. The realism index can be either defined internally, with the use of the prior latent density, or by some external feature space, e.g. similarly to Fréchet Inception Distance (FID) score (Heusel et al., 2017). In addition, we propose a simple to implement iterative algorithm, that optimises an interpolation with respect to the introduced index. As a consequence of our approach, we obtain an interpolation which simultaneously tries to optimise the two following features:

- the interpolating curve goes through regions where the generated points are realistic,
- the length of the curve transported to the input data space is small.

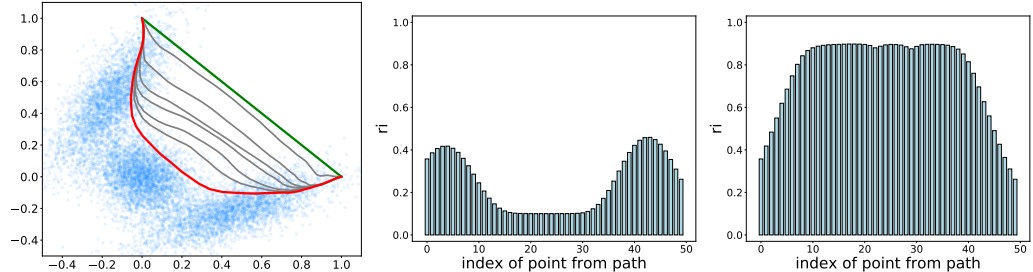

Figure 1: (left to right) Interpolation (projected onto two selected dimensions) for a semicircle distribution: green – linear, grey – consecutive interpolation steps, red – final proposed interpolation; values of the proposed reality index ri for consecutive points in the linear interpolation; values of ri for points in the optimised interpolation.

The above approach is especially valuable if either the prior density is not convex-shaped, or when a kind of a soap-bubble effect appears (i.e. when in the generative model, the points generated near the origin are of much worse quality than the ones chosen randomly).

## 2 REALISM INDEX OF A POINT

Let $X \subset \mathbb{R}^N$ be a dataset. Consider the case when we are given a manifold model for $X$, which consists of a decoder $\mathcal{G}$ from the latent $Z = \mathbb{R}^D$ to the input data space:

$$\mathcal{G} : Z \to \mathbb{R}^N. \tag{1}$$

This case covers both GAN-like and AE-like models.

We define the *realism index on $Z$* as a function

$$\mathrm{ri} : Z \to [0, 1] \tag{2}$$

such that high values of $\mathrm{ri}(z)$ indicate that $\mathcal{G}(z)$ is indistinguishable from the elements of $X$. In general, the optimal choice of ri can be nontrivial, and may depend on the generative model in question. In this paper we study two possible natural constructions. If the density $f$ in $Z$ is given, and its high values at point $z$ imply that $\mathcal{G}(z)$ is more realistic, we can base the construction of the realism index on the density. This assumption was empirically observed to be true for some generative models, such as the GLOW model (Kingma & Dhariwal, 2018). A different external approach can be constructed by using a separate network, which checks if a given point has similar features to that of the samples in the dataset (idea based on the Fréchet Inception Distance FID score).

**Normalisation procedure.** In all of our constructions we assume that we are given a function

$$f : Z \to \mathbb{R}_+$$

with higher values indicating more realistic points. To obtain the realism index based on $f$ we first need to apply the following normalisation procedure[1].

Let us first show the basic idea. Suppose that we have ordered the elements of the set $X$ (in the representation given from the latent $Z$) according to the order introduced by $f$. Then as the realism index of a given point $x$ we understand its normalised order in the sequence $f_{(i)}$. This motivates us to state the following definition.

**Definition 1.** *Let $\mathbb{X}$ be a random vector in $Z$, such that our data comes from the distribution $\mathcal{G}(\mathbb{X})$. We define the* realism index ri *based on the function $f$ by the formula*

$$\mathrm{ri}(z; f) := p(f(\mathbb{X}) \leq f(z)), \tag{3}$$

*where $p$ denotes the probability.*

---

[1]In general, even if $f$ is bounded, the obvious normalisation given by $\mathrm{ri}(x) = f(x)/\max f$ would not work, see Appendix A, Remark 1.

Observe that $ri(z, f) = \int_{\{w: f(w) \leq f(z)\}} f(s) ds$. Clearly it is the PDF of random variable $f(\mathbb{X})$ at point $f(z)$.

The proposed index lies in the $[0, 1]$ interval, and attains the value of $1$ only for points where $f$ attains maximal value. If $f$ is clear from the context, for notational convenience we shall write $\mathrm{ri}(z)$ instead of $\mathrm{ri}(z; f)$.

To practically apply the above concept, we need to be able to tune it to different generative models. To achieve this, we choose a value $\varepsilon < 1$ and, similarly to the approach used in GLOW (Kingma & Dhariwal, 2018), we rescale ri

$$\mathrm{ri}_\varepsilon(z) = \varepsilon + (1 - 2\varepsilon) \cdot \mathrm{ri}(z), \tag{4}$$

so that $ri_\varepsilon(z) \in [\varepsilon, 1 - \varepsilon]$. See Sec. 4 for further discussion on this rescaling.

**Realism index based on the normal density.** We shall now discuss the realism index based on the standard normal density $f = \mathcal{N}(0, I)$ in the $D$-dimensional latent $Z$. Let us choose a point $z$ from the latent and let $\mathbb{X}$ denote the random vector with $f$ density. Note that almost all points generated from the Gaussian density in $\mathbb{R}^D$ lie inside the sphere $S(0, D^{1/2})$. Points outside of this sphere can be considered unrealistic, while those inside might be obtained with random sampling.

We want to compute the probability

$$\mathrm{ri}(z) = p(f(\mathbb{X}) \leq f(z)). \tag{5}$$

By some easy calculations we get (see Appendix A for more details)

$$\mathrm{ri}(z) \approx \frac{1}{2} + \frac{1}{2}\mathrm{erf}\left(\sqrt{D - \frac{1}{2}} - \|z\|\right). \tag{6}$$

We can see that the density based realism index correctly identifies the latent points that lay inside the sphere $S(0, D^{1/2})$ as having an index approximately $1$, and points outside sphere as having index approximately $0$. Observe, that this behaviour is not recognised by the density itself, which has no clear change at the border of the sphere $S(0, D^{1/2})$.

**Realism index based on the Fréchet Inception Distance (FID).** If the density is either not available or not completely reliable, we can base the realism index on an external measure of sample credibility. To that end, we use an approach inspired by the Fréchet Inception Distance (Heusel et al., 2017) (FID). To compute FID score we start with a pretrained Inception network $I$. The entire training set $T$ is passed through $I$ to obtain a set of feature vectors, and we estimate its density $\mathcal{N}(\mu_T, \Sigma_T)$ by computing its mean $\mu_T$ and covariance matrix $\Sigma_T$. We apply the similar procedure to the data $W$ generated by our model, and obtain the density $\mathcal{N}(\mu_W, \sigma_W)$. Then we compute the Fréchet distance between two normal densities by the formula

$$FID(\mathcal{N}(\mu_T, \Sigma_T), \mathcal{N}(\mu_W, \Sigma_W)) = \|\mu_T - \mu_W\|_2^2 + Tr(\Sigma_T + \Sigma_W - 2(\Sigma_T \Sigma_W)^{1/2}).$$

However, in contrast to the original definition, we aim to compute the credibility of a single generated point, not the distance between two distributions. To achieve that goal, we use simply the likelihood of the point $x$ transported through $I$ with respect to density $\mathcal{N}(\mu_T, \Sigma_T)$:

$$f(x) = \mathcal{N}(\mu_T, \Sigma_T)(I(x)),$$

and apply $\mathrm{ri}_f$ as the constructed realism index. Observe that this realism index is based also on the density, but not in the latent space itself, but in some feature space constructed with the use of external network $I$.

**Numerical estimation of the realism index.** Clearly, for an arbitrary function $f$ the realism index does not have a closed form. In order to obtain differentiable estimation of ri, we draw sample $W = (w_i)_i$ from the random variable $\mathbb{X}$ (or simply choose it from the dataset), and compute values of $(f(w_i))_i$. Since the considered values are non-negative, to estimate the density we first proceed by logarithm to whole of $\mathbb{R}$ by taking $l_i = \log f(w_i)$, and compute either kernel or GMM density estimation $g$ of the random variable $\log(f(\mathbb{X}))$. Finally, we obtain that the estimator of the realism index $\mathrm{ri}(z; f)$ is given with the use of the cumulative density function of $g$: $\mathrm{ri}(z; f) \approx \mathrm{cdf}_g(\log f(z)))$.

## 3 REALISM INDEX OF AN INTERPOLATION

Our concept for the definition of the realism index for a path is inspired by transition between movie frames. The interpolation may be viewed as set of frames,where the first frame denotes the beginning of the path and last its end. Interpreting realism index as a probability that a given frame is realistic, we can define the respective index of the curve as the product of all its points.

**Realism index for naturally parameterised curves.** Let $\gamma : [0, T] \to Z$ be an interpolating curve, such that

$$\gamma(0) = x \text{ and } \gamma(T) = y, \tag{7}$$

for some given $x, y \in Z$. Additionally, we assume that the $\mathcal{G}\gamma$ is naturally parameterised

$$\|(\mathcal{G}\gamma)'(t)\| = 1 \text{ for } t \in [0, T]. \tag{8}$$

We discretize the curve by fixing a time-step $T/k$ (where $k$ denotes the number of frames) and consider the sequence of intervals $[\gamma(0), \gamma(T/k)], \ldots, [\gamma(T - T/k), \gamma(T)]$. To obtain the reality measure of this sequence we compute the product of all realisticity values of its points to the power equal to their "duration" $T/k$

$$\text{ri}(\gamma(t_0))^{T/k} \cdot \ldots \cdot \text{ri}(\gamma(t_k))^{T/k}, \tag{9}$$

where $t_i$ are the arbitrarily chosen intermediate points in the intervals $[iT/k, (i+1)T/k]$. By taking the logarithm of the above expression and proceeding with $k \to \infty$ we get

$$\sum_{i=1}^{k} \log \text{ri}\, \gamma(t_i) \cdot \tfrac{T}{k} \to \int_0^T \log \text{ri}(\gamma(t)) dt \text{ as } k \to \infty. \tag{10}$$

Consequently, we introduce the realism index of a naturally parameterised curve $\gamma : [0, T] \to \mathbb{R}^N$ by the formula

$$\text{ri}(\gamma) = \exp\left(\int_0^T \log \text{ri}(\gamma(t)) dt\right) \in [0, 1]. \tag{11}$$

Since every curve can be uniquely naturally parameterised, we interpret its index as the index of its natural reparameterisation. Therefore we arrive at the following general definition.

**Definition 2.** *Let* ri *be given realism index in $Z$. For an arbitrary curve $\gamma : [0, T] \to Z$ we define the* realism index ri *of $\gamma$ with*

$$\text{ri}(\gamma) = \exp\left(\int_0^T \log \text{ri}(\gamma(t)) \|(\mathcal{G}\gamma)'(t)\| \, dt\right) \in [0, 1]. \tag{12}$$

We further prove that the realism index of a curve is equal to its length with respect to a certain Riemann structure on the latent space. We will utilise this result in the next section in order to connect the search of optimal interpolation to the search of geodesics. Directly from the definition we get the formula for the realism index in terms of the latent

$$
\begin{aligned}
-\log \text{ri}(\gamma) &= -\int_0^T \log \text{ri}(\mathcal{G}\gamma(t)) \cdot \|(\mathcal{G}\alpha)'(t)\| dt = -\int_0^T \log \text{ri}(\gamma(t)) \cdot \sqrt{\langle (\mathcal{G}\gamma)'(t), (\mathcal{G}\gamma)'(t) \rangle} dt \\
&= -\int_0^T \log \text{ri}(\gamma(t)) \sqrt{\gamma'(t)^T [d\mathcal{G}(\gamma(t))]^T d\mathcal{G}(\gamma(t)) \gamma'(t)} dt \\
&= \int_0^T \sqrt{\log^2 \text{ri}(\gamma(t)) \gamma'(t)^T [d\mathcal{G}(\gamma(t))]^T d\mathcal{G}(\gamma(t)) \gamma'(t)} dt,
\end{aligned}
$$

where $d\mathcal{G}(x)$ denotes the derivative of $\mathcal{G}$ at point $x$. Consequently, we obtain the following theorem:

**Theorem 1.** *Let the Riemann structure in the latent space $Z$ be defined with the local scalar product $\langle, \rangle_z$ at a point $z$ using the following formula*

$$\langle v, w \rangle_z = v^T A_z w \text{ where } A_z = \log^2(\text{ri}(z)) d\mathcal{G}(z)^T d\mathcal{G}(z). \tag{13}$$

*Then*

$$\text{ri}(\gamma) = \exp(-\text{length}(\gamma; \langle, \rangle_z)), \tag{14}$$

where $length$ is the number of points in a path.

## 4 OPTIMAL INTERPOLATION

Considering the results from the previous section, we are able to formulate the definition of an ri-optimal curve.

**Definition 3.** *Let* ri *be a realism index in $Z$ for a generative model $\mathcal{G} : Z \to \mathbb{R}^N$. Let $x, y \in Z$ be fixed. We call a curve $\gamma : [0, T] \to Z$ such that $\gamma(0) = x$, $\gamma(T) = y$ ri-optimal (or, shortly, optimal) interpolation if it has the maximal realism index from all curves joining $x$ with $y$.*

Study first the issue of searching for, at least locally, optimal interpolations. Theorem 1 allows us to reformulate the problem as a task of finding geodesics. Consequently, the standard results from Riemann geometry apply (see Spivak (1999, Chapter 9)). Without loss of generality we can reduce the problem to the case $T = 1$ and optimise the length functional. However, due to the uniqueness of the local minima and local convexity of the functional, we can minimise the energy functional instead

$$E = \tfrac{1}{2} \int_0^1 \langle d\mathcal{G}(\gamma(t))\gamma'(t), \mathcal{G}(\gamma(t))\gamma'(t) \rangle_{\gamma(t)} dt. \tag{15}$$

The additional advantage is that curves which minimise the energy functional are parameterised proportionally to the natural parameterisation. Concluding, by applying Theorem 1, the optimal curve $\gamma : [0, 1] \to Z$, $\gamma(0) = x, \gamma(1) = y$, with respect to the realism index minimises:

$$E_x^y(\gamma) = \tfrac{1}{2} \int_0^1 \log^2(\mathrm{ri}(\gamma(t))) \| d\mathcal{G}(\gamma(t))\gamma'(t) \|^2 dt. \tag{16}$$

In general, the search for geodesics can lead to nontrivial computations involving second derivatives. However, for some special cases, we can significantly simplify the minimisation process. To justify this claim, we first introduce the formula for the discretization of the integral in the energy functional. Let $\gamma : [0, 1] \to \mathbb{R}^N$ be a curve such that

$$\gamma(0) = x \text{ and } \gamma(1) = y, \tag{17}$$

and divide the interval $[0, 1]$ into $k$ equal sub-intervals, and denote the values

$$\gamma(i/k) = x_i \text{ for } i = 0, \dots, k. \tag{18}$$

For $k - 1$ vectors $x_1, \dots, x_{k-1}$ in $\mathbb{R}^N$ (where $x_0 = x$, $x_k = y$), approximate the value of (16) with

$$
\begin{aligned}
2E_x^y(\gamma) &\approx \sum_{i=0}^{k-1} \log^2 \tfrac{\mathrm{ri}(\gamma((i+1)/k)) + \mathrm{ri}(\gamma(i/k))}{2} \cdot \left( \tfrac{\|\gamma((i+1)/k) - \gamma(i/k)\|}{1/k} \right)^2 \cdot \tfrac{1}{k} \\
&= k \log^2 \tfrac{\mathrm{ri}(x) + \mathrm{ri}(x_1)}{2} \|x_1 - x\|^2 + k \sum_{i=1}^{k-2} \log^2 \tfrac{\mathrm{ri}(x_i) + \mathrm{ri}(x_{i+1})}{2} \|x_{i+1} - x_i\|^2 \\
&\quad + k \log^2 \tfrac{\mathrm{ri}(x_{k-1}) + \mathrm{ri}(y)}{2} \|y - x_{k-1}\|^2.
\end{aligned}
$$

Considering all the above computations, to compute an optimal interpolation we need to minimise

$$
\begin{aligned}
\tfrac{2}{k} \cdot E_x^y(x_1, \dots, x_{k-1}) = &\log^2 \tfrac{\mathrm{ri}(x) + \mathrm{ri}(x_1)}{2} \|x_1 - x\|^2 + \sum_{i=1}^{k-2} \log^2 \tfrac{\mathrm{ri}(x_i) + \mathrm{ri}(x_{i+1})}{2} \|x_{i+1} - x_i\|^2 \\
&+ \log^2 \tfrac{\mathrm{ri}(x_{k-1}) + \mathrm{ri}(y)}{2} \|y - x_{k-1}\|^2
\end{aligned} \tag{19}
$$

over $x_1, \dots, x_{k-1} \in \mathbb{R}^N$.

**Optimisation procedure.** In all the experiments in the paper we achieve this goal using the standard gradient descent method initialising $x_i$ with a linear interpolation $x_i = \left(1 - \frac{i}{k}\right) x + \frac{i}{k} y$ for $i = 1, \dots, k - 1$. However, to accelerate the process of minimisation[2], we alternate the gradient step

---

[2]This step is essential when the initial interpolation passes through regions with density close to zero, which could result in a vanishing gradient problem, see Appendix C.

with the following one: first choose two random numbers $i < j$ from the set $\{0, \ldots, k\}$ and then consider the linear interpolation between $x_i$ and $x_j$ given with

$$\bar{x}_l = \tfrac{j-l}{j-i}x_i + \tfrac{l-i}{j-i}x_j \text{ for } l \in \{i+1.\ldots, j-1\}.$$

Finally, if the linear interpolation has smaller energy then the original part, i.e. when

$$E_{x_i}^{x_j}(\bar{x}_{i+1}, \ldots, \bar{x}_{j-1}) < E_{x_i}^{x_j}(x_{i+1}, \ldots, x_{j-1}),$$

we replace $x_l$ by $\bar{x}_l$ for $l = \{i+1, \ldots, j-1\}$.

**Effects of $\varepsilon$-regularisation on optimisation.** Given a reality index ri$(\cdot)$, see eq. (4), we have introduced its regularisation ri$_\varepsilon(z) = \varepsilon + (1-2\varepsilon)\cdot$ri$(z)$, $\varepsilon \in [0, 1/2]$. In practice, in all experiments we typically choose $\varepsilon = 0.1$. The reason behind such regularisation is twofold. First, if $\varepsilon = 0$ and ri is zero (or numerically close to zero) on some subset of the domain (for example if the density is zero), the optimisation for points with initial interpolating interval there is unmanageable (observe that in the realism of the curve we take the logarithm of the index). The case when $\varepsilon = 0$ and ri $= 1$ at some subset of the domain can also cause problems, as then logarithm of the reality index in this set is zero, and consequently the interpolating curve has no cost of staying or going through this region. Consequently, in our experience it is best to regularise ri by restricting its image to a subset of interval $[\varepsilon, 1 - \varepsilon]$.

Let us now discuss the special limiting case when $\varepsilon = 1/2$. Then clearly

$$\text{ri}_\varepsilon \equiv 1/2.$$

Let us first recall that we are given a decoder (generator) $\mathcal{G} : Z \to \mathbb{R}^N$, such that the data set lies in the manifold given by $M = \mathcal{G}(Z)$. Let us consider the interpolating curve $\gamma : [0, 1] \to Z$. Observe that in that case the energy function does not depend on ri and equals

$$E_x^y(\gamma) = \tfrac{1}{2} \int_0^1 \log^2(\text{ri}_\varepsilon(\gamma(t))) \|d\mathcal{G}(\gamma(t))\gamma'(t)\| dt$$

$$= \tfrac{1}{2} \log^2(2) \int_0^1 \|d\mathcal{G}(\gamma(t))\gamma'(t)\| dt = \tfrac{1}{2} \log^2(2) \cdot \text{length}(\mathcal{G} \circ \gamma).$$

Thus the minimal $E_x^y(\gamma)$ value would be obtained for a curve joining $x$ with $y$ whose length measured in the input space is minimal. Thus if $\bar{x} = \mathcal{G}(x)$ and $\bar{y} = \mathcal{G}(y)$ are points in the input space, then the interpolating curve with minimal energy is equal to one that connects points $\bar{x}$ and $\bar{y}$ with a curve in the manifold $M$ which has minimal length (measured in input space). Consequently, although the limiting case does not take into account the realism index of the interpolation, it still will usually produce interpolations comparable to the common linear. See Appendix D for additional analysis.

## 5 EXPERIMENTS

In all the considered experiments we apply the regularised version or realism index ri$_\varepsilon$ with $\varepsilon = 0.1$. First we are going to consider the case when the index is based on the prior density in the latent, and next we briefly discuss the realism index based on the FID score.

**Density based index.** In this part we demonstrate our method's ability to produce more meaningful interpolations. In order to achieve this goal we use a DCGAN model (Radford et al., 2015) trained on MNIST and Celeb-A datasets (LeCun et al., 1998; Liu et al., 2015). We consider a non-trivial latent created from a conjunction of three multidimensional Gaussian distributions

$$p(z) = \tfrac{1}{3} \sum_{i=1}^3 \mathcal{N}(\mu_i, \Sigma_i)(z) \text{ for } z \in Z, \tag{20}$$

where $\mu_i := (\tilde{\mu}_i, 0, \ldots, 0)^T \in \mathbb{R}^{20}$ for $\tilde{\mu}_1 = (2, 6)$, $\tilde{\mu}_2 = (0, 0)$, $\tilde{\mu}_3 = (2, -6)$ and $\Sigma_i := \begin{bmatrix} M_{11}^{(i)} & M_{12} \\ M_{21} & M_{22} \end{bmatrix}$ for $M_{12} = M_{21} = \mathbf{0}$, $M_{22}$ is 2-D array with 0.5 on the diagonal and zeros elsewhere, and $M_{11}^{(1)} = \begin{bmatrix} 5 & 2 \\ 2 & 2 \end{bmatrix}$, $M_{11}^{(2)} = \begin{bmatrix} 1 & 0 \\ 0 & 3 \end{bmatrix}$, $M_{11}^{(3)} = \begin{bmatrix} 5 & -2 \\ -2 & 2 \end{bmatrix}$. From here on we shall call it a semicircle, see Fig. 1 (leftmost subfigure).

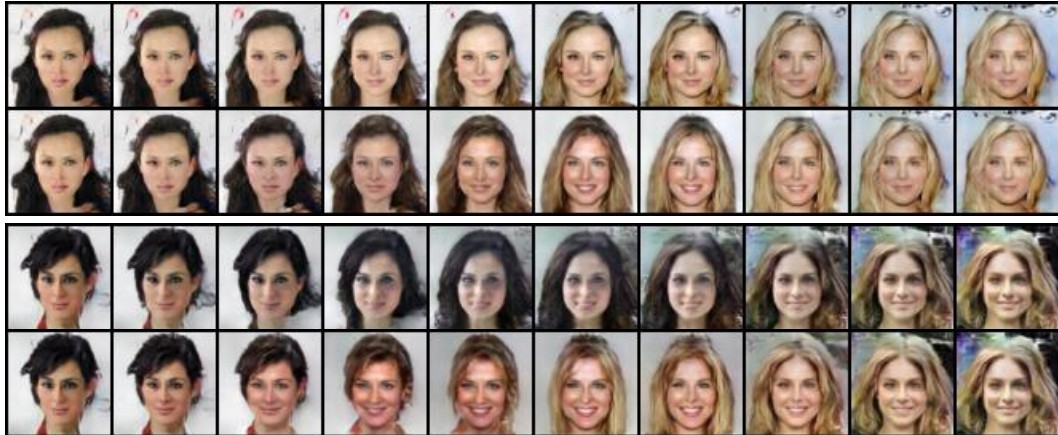

Figure 2: Examples of results for linear interpolation (top row in each example) and our results (bottom row in each example) for Celeb-A dataset, with a semicircle prior trained DCGAN. Observe that since in the linear interpolation the middle point is far from the density of the data, we can often observe in it some artefacts.

Results of sampling from a GAN trained with the semicircle prior are shown in Fig. 2 with respective projection shown in Fig. 1. We randomly choose two points from different ends of the distribution and start the algorithm's minimisation procedure given with eq. (19) with a linear interpolation (green line in Fig. 1). It is obvious that the midpoints of initial path are sampled from a very low-density areas, completely outside the prior.

In Fig. 1 you can also the ri index values for the linear (center) and proposed (rightmost image) interpolations. Our method allows for more dynamic behaviour in this matter, especially at the end points. Images in the path obtained with proposed algorithm differ more, and at the same time seem more real.

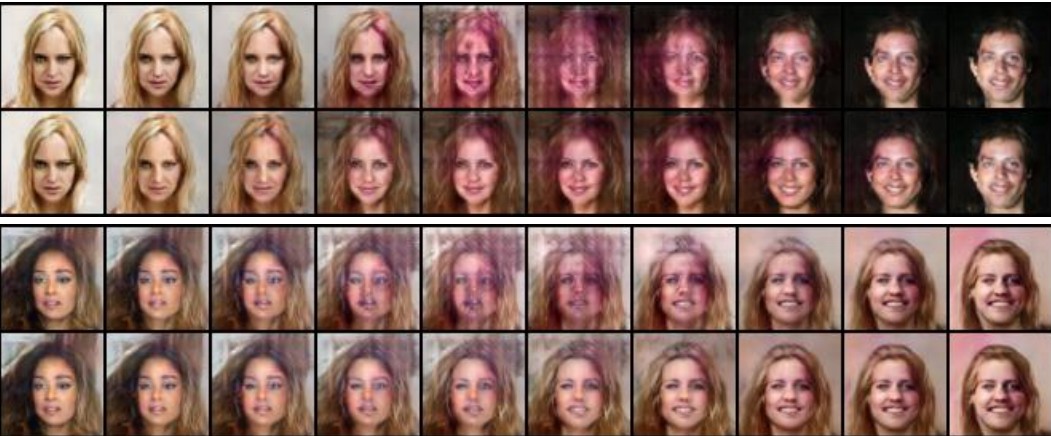

Figure 3: Examples of results for linear interpolation (top rows of interpolation pairs) and our FID-score based results (bottom rows of interpolation pairs) for Celeb-A dataset. The interpolation was optimised had 50 midpoints, and we show here 10 of them.

**FID-score based realism index.** In order to test our FID-score based realism index, we perform experiments on a GAN, trained on the Celeb-A dataset. Since samples from a high dimensional Gaussian approximately cluster on a sphere, points in the latent space that are very close to the origin are virtually never seen during training. Therefore, samples generated in that area may or may not be realistic. We selected a GAN model, for which we noticed that decoding latent codes near the origin gives low quality samples (the earlier mentioned soap-bubble effect), which makes

for a perfect test environment for the FID-score based realism index. We aim to choose interpolation endpoints $x$ and $y$ in such a way, that the linear interpolation between them would pass close to the origin in the latent space. To that end, we first sample $x$ uniformly, and then produce $y$ from $x$ by multiplying by $-1$ a random subset of dimensions.

The results of these experiments can be seen in Fig. 3. We can see that the linear interpolations yield very low quality samples (especially in the middle of the interpolation). After optimising the linear paths using the FID-score realism index, the paths avoid unrealistic regions in the latent space, producing samples of much higher quality. Note that this experiment does not require access to the density. Instead, it is based *only* on the training samples.

Even though FID score has some limitations (Shmelkov et al., 2018), observe that its application in our model results in more realistic interpolations, as compared to linear ones. Use of a measure more adapted to evaluating quality of data could result in further improvement in the realism of interpolations.

**Further discussion.** In this part we discuss when the linear approximations will be close to those given by linear. Let us first consider the case when the generator is linear with prior uniform distribution in the latent.

**Observation 1.** *Let $U$ be a convex bounded set in $Z$ and let $f$ denote the uniform distribution on $U$. We consider the case of linear generative model, i.e. where $\mathcal{G}$ is a linear map $x \to Ax$ (with $A$ injective). Then the ri-optimal interpolations are given by the linear interpolations.*

*Proof.* Clearly, we can equivalently compute the realism index of a curve $\gamma$ connecting two points $x, y$ by computing the standard euclidean distance of $A\gamma$ in the convex set $AU$. More precisely: $\mathrm{ri}(\gamma; \mathrm{uni}_U) = \exp(-\mathrm{length}(A\gamma))$. Since linear maps move intervals onto intervals, we obtain the assertion of the observation. $\square$

In practice, similar behaviour happens when the derivative of the generator has small variation and the realism index is close to being a constant one. Observe, that by equation (6), for a generative model with high dimension of the latent space and Gaussian prior, the realism index is almost constant on a linear interpolation of arbitrary randomly chosen points (except for a possibly small neighbourhood of the endpoints). Consequently, when the derivative of generator $\mathcal{G}$ does not vary too much in the vicinity of the linear interpolation, the linear interpolation will be close to optimal find by our approach. In practice, we observe this behaviour in auto-encoder based generative

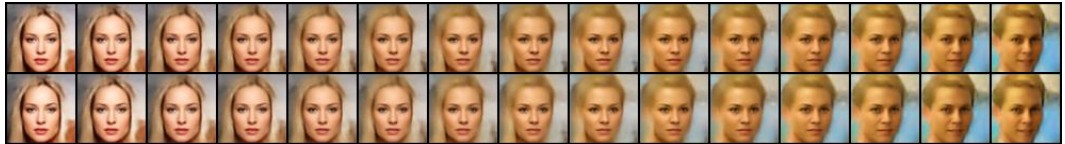

Figure 4: Linear (top) and our interpolation (bottom) in the WAE (Tolstikhin et al., 2017) model. Observe that the consecutive elements of the interpolations are practically identical.

models, such as Wasserstein Auto-Encoders, see Fig. 4. This follows from the fact that too high variation (of derivative) of the decoder is penalised (regularised) by its approximate inverse given by the encoder.

## 6 CONCLUSIONS

In this paper we studied the problem of generating a meaningful interpolation from a previously trained generative model, either a GAN or a generative auto-encoder. We claim that a good interpolation should both reveal the hidden structure of the dataset, as well as be smooth and follow the true data distribution, i.e. produce realistic elements.

In order to produce curves satisfying these conditions we define a *realism* index of a path, which takes into account both density values and and differences between consecutive decoded images to ensure smoothness. We show how to define realism index using either a known density or, in case

it is either not available or not reliable, how to base it on some external measure, e.g. the FID score. We have proved that this interpolation procedure is equal to finding a geodesics with reality index equal to its length in respect to some latent space Riemann structure.

For the practical use, we have defined the notion of an optimal interpolation, and proposed a simple and efficient numerical procedure for its search. The experiments show that the constructed interpolations are in superior to the linear ones, making it possible to escape regions of low data density or low data quality, both for the density- and FID-based approaches. This is especially visible if the prior density is not Gaussian, when the linear interpolations often proceed through regions in space of extremely low density. Another example when the linear interpolations are suboptimal to our method is given by the standard GAN model when the soap bubble effect appears.

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

# A  CASE OF THE NORMAL DENSITY.

We shall now compute the probability index of the standard normal density $f = \mathcal{N}(0, I)$ in the $D$-dimensional latent $Z$. Let us choose a point $z$ from the latent and let $\mathbb{X}$ denote the random vector with density $f$. We want to compute the probability

$$\mathrm{ri}(z) = p(f(\mathbb{X}) \leq f(z)). \tag{21}$$

Observe that from the definition of normal density we have

$$p(f(X) \leq f(z)) = p(\|\mathbb{X}\|^2 \geq \|z\|^2) = 1 - p(\|\mathbb{X}\|^2 \leq \|z\|^2). \tag{22}$$

Since $\|\mathbb{X}\|^2$ has the chi-square distribution with $D$ degrees of freedom, we obtain that

$$\mathrm{ri}(z) = 1 - F(\|z\|^2; D), \tag{23}$$

where $F(r; D)$ denotes the cumulative distribution function of the chi-square density $\chi^2(D)$ (with $D$ degrees of freedom). Let us now proceed with an asymptotic analysis. Observe that

$$p(\|\mathbb{X}\|^2 \leq \|z\|^2) = p(\sqrt{2}\|\mathbb{X}\| \leq \sqrt{2}\|z\|). \tag{24}$$

If $\mathbb{Y} \sim \chi^2(D)$ then for large $D > 30$, $\sqrt{2\mathbb{Y}} - \sqrt{2D - 1}$ is approximately normally distributed (see Johnson et al. (1994, formula (18.23) on p. 426)). Consequently

$$\mathrm{ri}(z) = 1 - p(\sqrt{2}\|\mathbb{X}\| \leq \sqrt{2}\|z\|) \approx 1 - \Phi(\sqrt{2}\|z\| - \sqrt{2D - 1}). \tag{25}$$

where $\Phi$ denotes cdf of standard Gaussian. Since $\Phi(r) = \frac{1}{2}\left[1 + \mathrm{erf}(r/\sqrt{2})\right]$ we get

$$\mathrm{ri}(z) \approx \frac{1}{2} + \frac{1}{2}\mathrm{erf}\left(\sqrt{D - \frac{1}{2}} - \|z\|\right). \tag{26}$$

**Remark 1.** *The above formula implies that for the normal density $f = N(0, I)$ in $\mathbb{R}^D$ the realism index is approximately $1$ in the ball $B(0, r_D - 3)$ and approximately $0$ outside of the ball $B(0, r_D + 3)$, where $r_D = \sqrt{D - \frac{1}{2}}$. This behaviour is natural and expected, as most of the points generated from normal density concentrate around the sphere $S(0, r_D)$.*

*Let us now discuss why we use the normalisation given by eq. (4), and not the seemingly natural given for bounded $f$ by the formula*

$$\mathrm{ri}(z) = \frac{f(z)}{\max f}.$$

*Consider the case when $f = \mathcal{N}(0, I)$ in $\mathbb{R}^D$. Then $\mathrm{ri}(z) = \exp(-\|x\|^2/2)$, which means that $\mathrm{ri}(0) = 1$, but $\mathrm{ri}(z) = \exp(-D/2 + 1/4)$ for $z \in S(0, r_D)$. Since the randomly generated point from $N(0, I)$ has norm close to $r_D$, this implies that almost every point which comes from the normal density would have realism index close to $0$, which would be of undesired and pathological behaviour.*

In practice, we observe this behaviour in auto-encoder based generative models, such as Wasserstein Auto-Encoders, see Fig. 4. This follows from the fact that too high variation (of derivative) of the decoder is penalised by its approximate inverse given by the encoder.

# B  DENSITY BASED REALISM INDEX IN GAN MODELS

In this section we present additional results for the optimisation of the density based realism index of an interpolation curve in a GAN model. The experiments are conducted on the Celeb-A and MNIST datasets, using the DCGAN architecture. The optimisation procedure is initialised with a linear interpolation consisting of 50 points and implemented using *Adam* optimiser. To show the relation of the starting linear interpolation to the one obtained from the proposed procedure, see e.g. Fig. 1, we perform a projection of $k$ interpolation $z_i$ points onto point $(x, y) \in \mathbb{R}^2$ such that

$$x \cdot z_0 + y \cdot z_k = \tilde{z}_i \quad \text{for} \ \ i = 0, \dots, k, \tag{27}$$

where $\tilde{z}_i := X(X^T X)^{-1} X^T z_i$ are the latent space points $z_i \in Z$ and $X = [z_0, z_k]$.

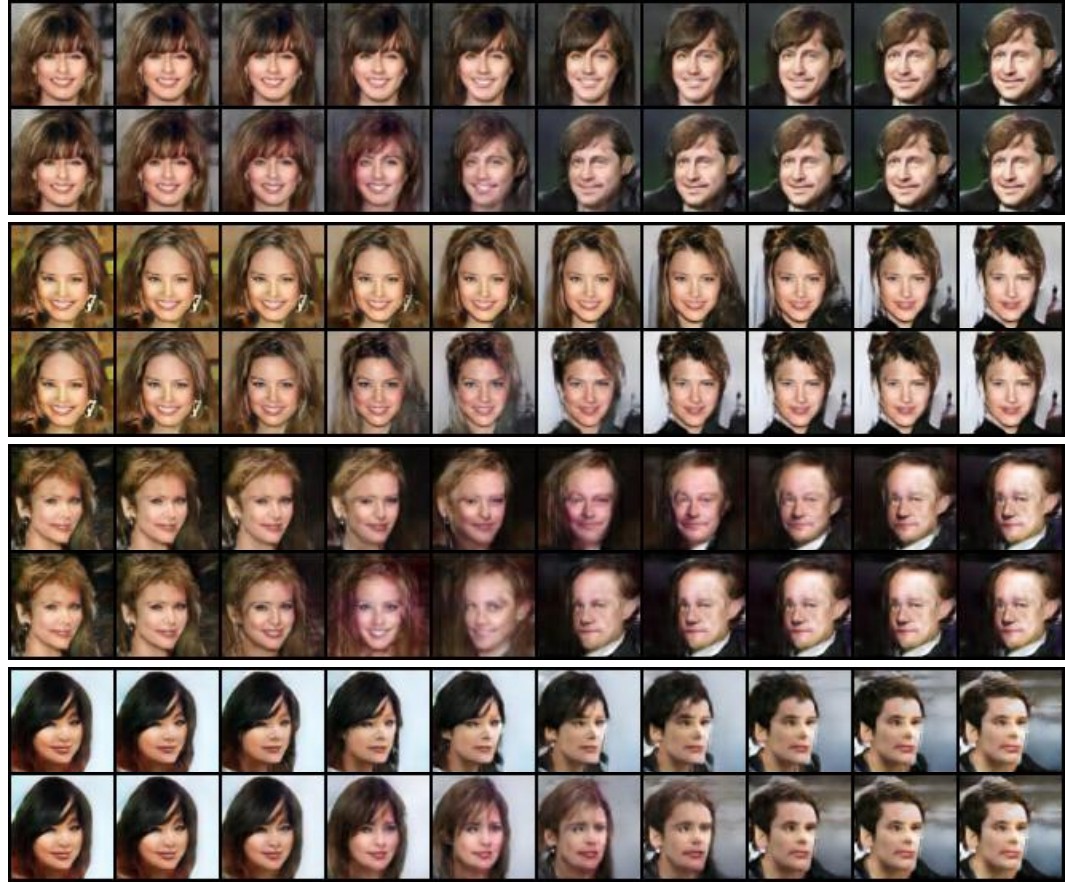

Figure 5: Interpolation points in DCGAN model trained with normal prior. In each pair the top represents the linear path and the bottom shows the results of our optimisation procedure. Each line consists of 10 equally spaced images selected from the 50 points that form the path.

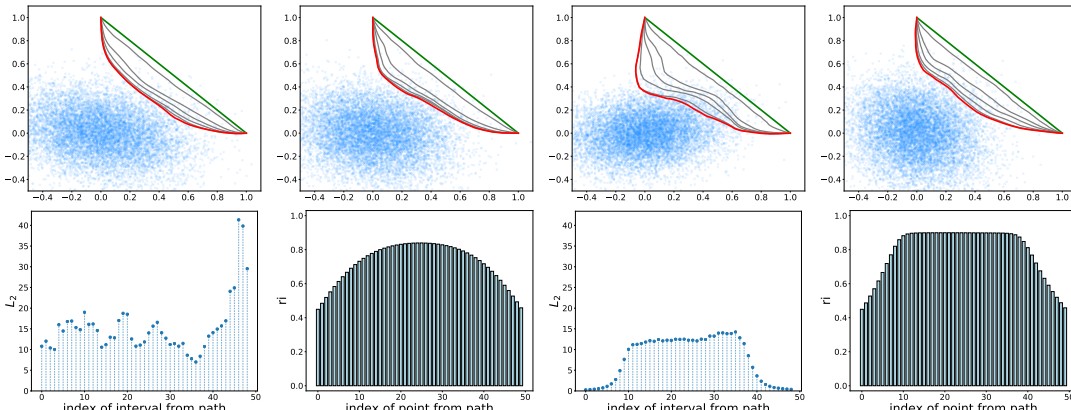

Figure 6: **Top:** Projections (see (27)) of the latent sample's density (blue dots), linear (green line) and proposed (red line) interpolations shown in Fig. 5 from top to bottom, respectively. **Bottom:** The squared $L_2$ distances between consecutive points and the realism index ri of each point in the path from the last example in Fig. 5. *Left*: The initial linear path. *Right*: The path at the end of optimisation procedure.

The resulting images for Celeb-A for the Gaussian prior are shown in Fig. 5, and Fig. 6 shows their path projections. It can be easily noticed that the proposed interpolation gives more dynamical objects. Also the interpolation is pulled by the density. Similar observations can be observed for

semicircle prior. The resulting images for Celeb-A together with their path projections for semicircle prior are shown in Fig. 7. We made also the same experiments for MNIST dataset. The resulting images for this data for the Gaussian and semicircle prior are shown in Fig. 8.

In the case of auto-encoder based architectures we advice the use of the WAE model, since it obtains slightly better interpolations and reconstructions than VAE – see Fig. 9.

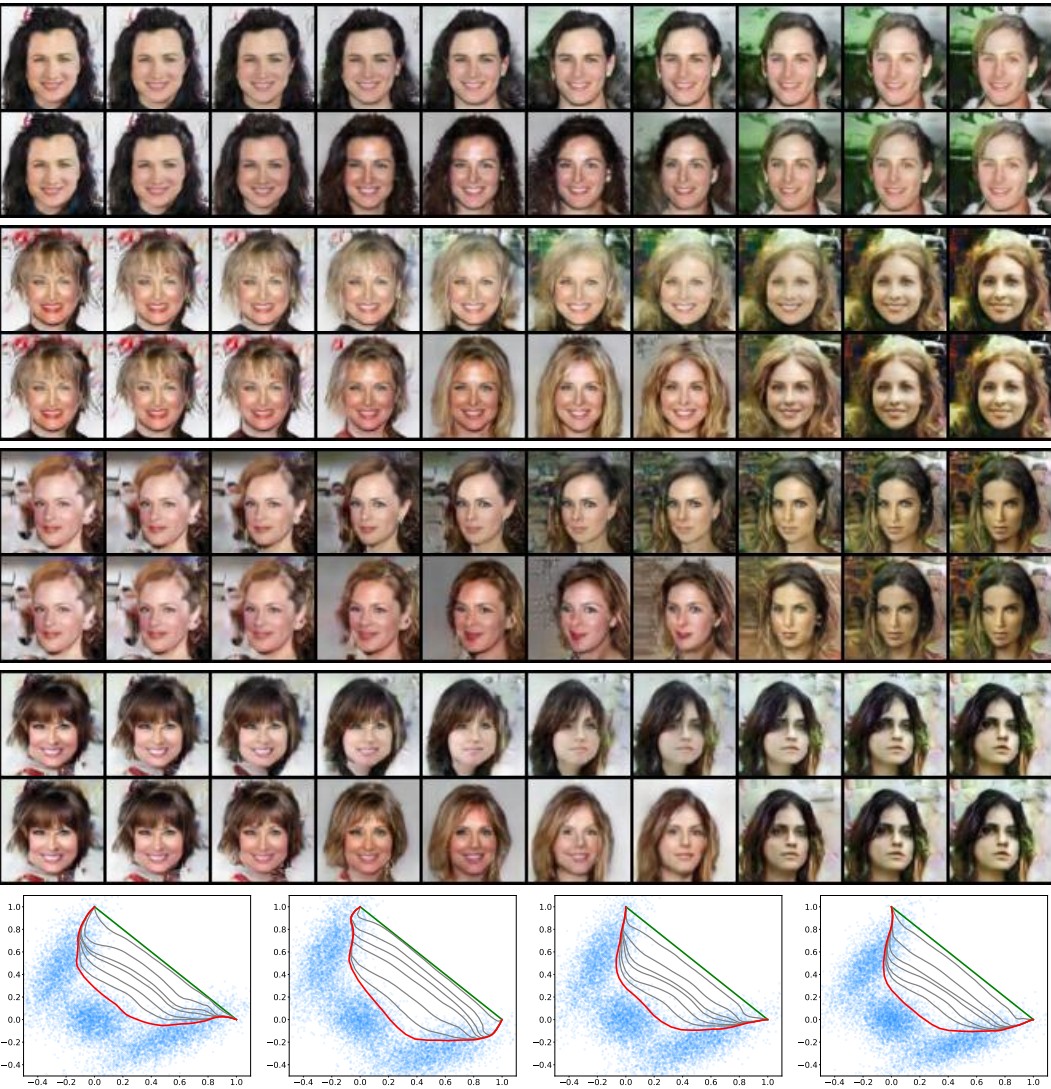

Figure 7: **Top:** Interpolation points in DCGAN model trained with semicircle prior. In each pair the top represents the linear path and the bottom shows the results of our optimisation procedure. Each line consists of 10 equally spaced images selected from the 50 points that form the path. **Bottom:** Selected projections of the latent sample's density (blue dots), linear (green line) and proposed (red line) interpolations shown in 'top' part together with the interpolation progress (in gray).

## C   OPTIMISATION PROCEDURE

In the performed experiments we use the optimisation described in Section 4 to reduce the number of necessary iterations in our method. In the initial phase of the optimisation of the linear interpolation, the value of the gradient for the middle points (located in the middle of the interpolation curve) can be extremely small due to the diminishing density. This is particularly evident for a DCGAN model with a semicircle prior, see Fig. 10. In consequence, this part of the curve converges slower than the points lying closer to the ends of the curve. Our experiments for a semicircle distribution show that

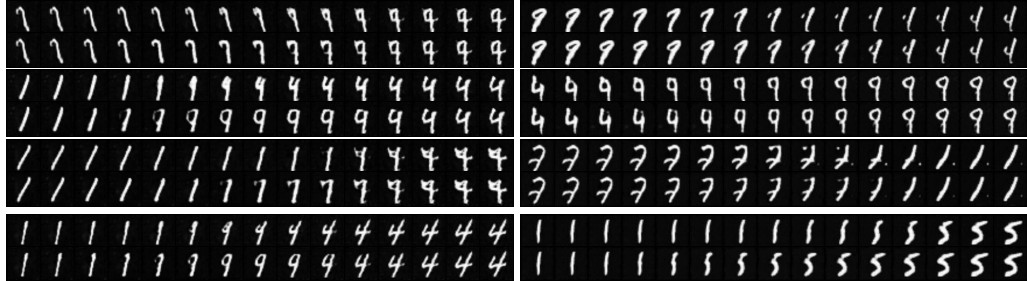

Figure 8: Linear and proposed interpolation paths for the MNIST dataset (a semicircle latent prior model in left column, a multidimensional normal in right column) in a DCGAN model.

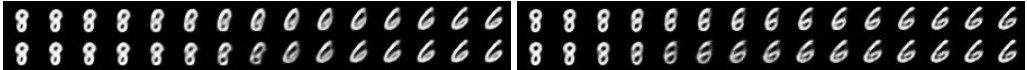

Figure 9: Comparison of interpolations in VAE (left) and WAE (right) models. The top line is the linear interpolation, the bottom one is the obtained with use of the density realism index approach.

we obtain the same final result without using the described acceleration, but with up to 3 times more iterations. This optimisation procedure is crucial only for non typical densities; for the classical normal prior no significant impact on the number of iterations is observed.

## D   THE $\varepsilon$ VALUE IMPACT ON THE INTERPOLATION CURVE

In Fig. 11 we present squared $L_2$ distances between consecutive points in the optimised path from the last example from Fig. 2. In this experiment we use different values of epsilon $\varepsilon = 10^{-5}, 10^{-4}, 10^{-3}, 0.01, 0.1, 0.25$ for $\mathrm{ri}_\varepsilon$ (see eq. (4)). As one can see, distances between consecutive points do not vary much for $\varepsilon$ values much larger than zero.

Higher $\varepsilon$ values result in more equally spaced interpolations, and the interpolation optimisation process is faster. However, higher $\varepsilon$ result in puts a different emphasis on terms of the formula optimised (see eq. (19) and discussion on $\varepsilon$ in Sec. 4), i.e. lengths start to be more important (hence the inter-point distances) than individual point realities.

On the other hand, lower $\varepsilon$ values result in non-equally spaced points on the interpolation, which is easily visible in Fig. 11, while the optimisation process is slower.

Empirically, we found $\varepsilon = 0.1$ to be optimal, and therefore it is used as default value. Experiments conducted on a Celeb-A dataset gave similar results.

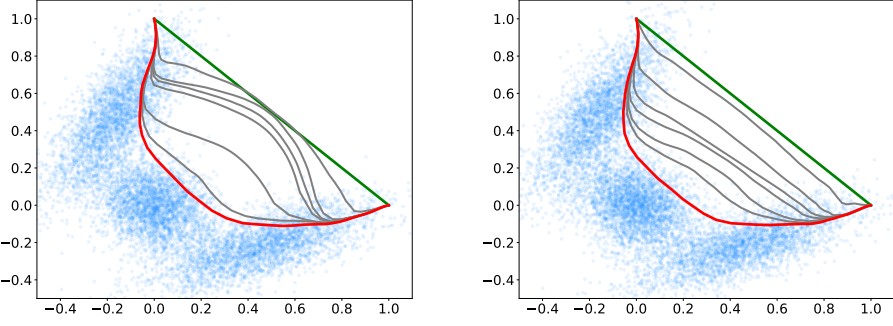

Figure 10: Interpolations (projected defines by (27)) for a semicircle distribution: green – linear, grey – consecutive interpolation steps, red – final proposed interpolation. The picture on the left shows result of our method without 'Optimisation procedure', and right side with optimisation.

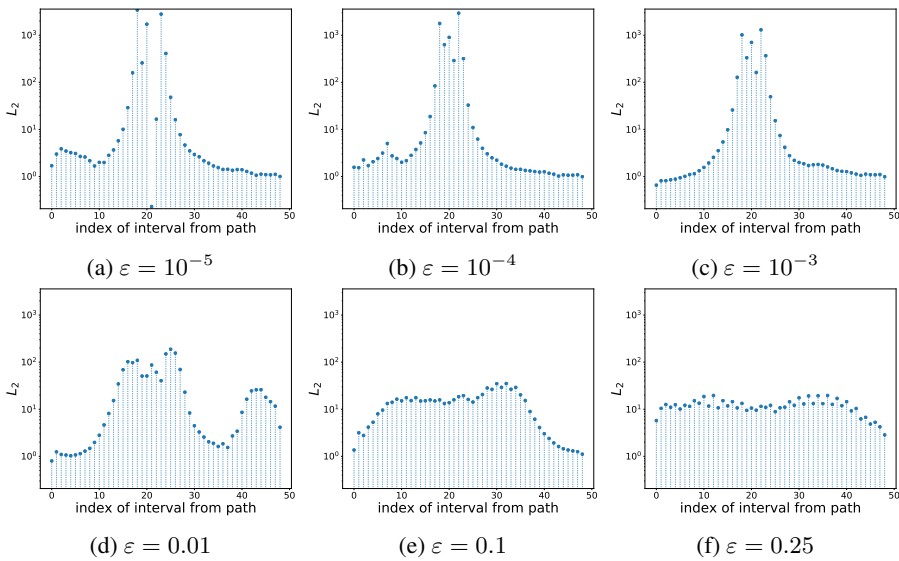

(a) $\varepsilon = 10^{-5}$  (b) $\varepsilon = 10^{-4}$  (c) $\varepsilon = 10^{-3}$

(d) $\varepsilon = 0.01$  (e) $\varepsilon = 0.1$  (f) $\varepsilon = 0.25$

Figure 11: The squared $L_2$ distances between consecutive points in the last example path from Fig. 2 for our interpolation and different values of $\varepsilon$ for $\mathrm{ri}_\varepsilon$ (see eq. (4)).

