# OpenReview forum: "Realism Index: Interpolation in Generative Models With Arbitrary Prior"
_ICLR.cc/2020/Conference — Reject_

### Official Review · AnonReviewer2 · 2019-10-23
**Official Blind Review #2**

**Rating:** 3

**Review:**

This paper introduced a linear interpolation method that could be applied to the latent space of a generative model.  With their method, interpolating instances generated by those generative models all maintain high quality in terms of the realism index they proposed.

This paper first introduced the quantity realism index, which is a measure of how well a generated instance in the latent space is fitted to the ground truth manifold of the data space. The definition of realism index is the probability measure of the sublevel set of some f (f:latent -> R+) function threshold by the f(z) for some z. f(z) could be the density function of feature in the latent space.

There are two types of realism index which has analytic form introduced in this paper. The one based on normal density. Another is based on Frechet inception distance. If the density of latent feature is normal, normal density based index is used and it could be approximated with a analytic form; while if the density is not accessible, then f is the gaussian density of certain transformation of features in the latent space.

If an arbitrary f is used, a kernel density estimator is used to estimate the density of log(f). After the realism index is introduced, the optimal interpolation  is the one has the highest cumulative realism index along the interpolation curve.

To optimize, a linear interpolation is used as initialization. All the intermediate results are updated iteratively.

Results show better interpolation than linear ones. But not many baselines are available.

Overall, I think the problem is very interesting and important. The results seem reasonable although only beating an obviously flawed baseline.

Typo in Equation (3): w->s?

-----------------

After reading rebuttal and other reviews, I agree that quantitative results is crucially missing especially when the proposed method involves proposing an optimization method to find the best geodesic. I think the authors should find a reasonable model that can at least show the proposed geodesic is better than a linear interpolation in the accumulated realism score. Also some quantitative evaluation of the proposed optimization scheme will be very helpful.

**Experience Assessment:**

I do not know much about this area.

**Review Assessment: Checking Correctness Of Derivations And Theory:**

I assessed the sensibility of the derivations and theory.

**Review Assessment: Checking Correctness Of Experiments:**

I assessed the sensibility of the experiments.

**Review Assessment: Thoroughness In Paper Reading:**

I made a quick assessment of this paper.

---

> ### Author Response · Authors · 2019-11-07
> **Thank you for your review**
>
> Thank you for reviewing our work! We believe that Equation (3) is correct, although the notation used there was not explained clearly enough. The letter “w” in this Equation is only used to define the set over which we are integrating. We are grateful to the reviewer for pointing out this issue, and we will update the paper to address it.

---

### Official Review · AnonReviewer4 · 2019-11-03
**Official Blind Review #4**

**Rating:** 3

**Review:**

The paper proposes a density score, which is defined based on a density or some other score such as Fréchet Inception Distance (FID).

While the given problem is an important one, there are some substantial problems with the proposed approach:
1) The paper does not mention the limitations of the FID which, as described in (Shmelkov et al, ECCV 2018. How good is my GAN?), are: i) it is based on the pre-trained Inception network and therefore does not exactly match the distributions over the data in other datasets; and ii) crude approximation of the scores by Gaussian distributions. Overall, this score is empirical and aims at circumventing the subjective analysis and it should be better reflected in the paper.
2) The justification of the normal density based index (Section 2) seems weak. While it is obvious that this score could be used, is it possible to make the empirical assessment? E.g. compare between two scores on an extensive amount of data.
3) In addition to the previous comment, there is a substantial problem in the experimental results that all the observations are qualitative and based only on a few images. Further analysis of the realism index on the real images which are not included into the training dataset could improve the analysis.
4) Following up on the previous comment, the experiments on the real images may need the values from the latent space corresponding to the real images. Currently, the model has been mostly assessed using DCGAN (with the assessment of VAE in Figure 4 but only for FID score), while it is stated after equation (1) that 'This case covers both GAN-like and AE-like models.'  This might be used for the assessment of the realism index on real images as stated before. If the proposed model were assessed with variational auto-encoders (VAE) and flow based models, it would make it possible to transform between the latent representation and the data themselves.  From another perspective, experimental results on different types of models (VAE-type, flow based type such as GLOW) for different types of interpolation are needed for the sake of experimental completeness. It would help emphasise the limitations of the method and difference in interpolation results in different models.
4) In the optimisation section, the following statement is made: 'However, to accelerate the process of minimisation, we alternate the gradient step with the following one: first choose two random numbers i < j from the set {0, . . . , k} and then consider the linear interpolation between xi and xj given with...'. Could the authors elaborate on why does this acceleration happen? It might be necessary to give some references on the experimental results or at least provide some line of support for this phrase.
***
The following comments are not as critical but fall into the category of ‘nice to have':
5) Although the reviewer is aware that there were some experiments in the appendix on the value \epsilon, it might be a good idea to have more studies on the influence of this regularisation parameter for other datasets rather than just MNIST
6)  While figure 1 appears in the beginning of the paper, on page two, it is discussed on page seven, in the experimental section. Placing the figures closer to the narrative would improve the reading experience.

**Experience Assessment:**

I have read many papers in this area.

**Review Assessment: Checking Correctness Of Derivations And Theory:**

I carefully checked the derivations and theory.

**Review Assessment: Checking Correctness Of Experiments:**

I carefully checked the experiments.

**Review Assessment: Thoroughness In Paper Reading:**

I read the paper at least twice and used my best judgement in assessing the paper.

---

> ### Author Response · Authors · 2019-11-07
> **Thank you for your review**
>
> Thank you for your very detailed comments and suggestions! Below we review each individual point that you made.
>
> 1) We agree that the FID score clearly has many limitations, and we are happy to expand the discussion about them in the paper. However, the focus of our work is not to discuss the pros and cons of FID score, but to present a new interpolation technique that could be based on an arbitrary score. It should be noted that despite the fact that FID score is empirical, its application as the realism score in our method still gives reasonable results. Exchanging FID score with a better realism measure would likely further improve the performance. The main issue with the measures GAN-train and GAN-test proposed by Shmelkov et al. is that they require a separate classification model, optimized either using the GAN training set (GAN-train) or examples generated from the GAN (GAN-test). Therefore, they are suitable for datasets with a clear classification task (and would be problematic, for instance, for the Celeb-A dataset), while our realism index can be used for any dataset. We agree that designing a good realism measure is a challenging problem, and we will update the paper to address your concerns.
>
> 2) Our paper includes a theoretical derivation of the approximation of the realism index in the case of the normal distribution (6), which we believe gives a strong basis for the method. The key point of this derivation is to discuss the properties of the proposed index. It demonstrates that the proposed realism index exhibits the expected behavior for high dimensional normal density:
> - As it is well known, almost all points generated from the standard normal density in R^D lie inside the sphere S(0,D^½). Thus points outside of this sphere can be considered unrealistic, while those inside can be obtained by random sampling.
> - By examining equation (6), we directly see that our realism density based index correctly identifies the latent points that lay inside the sphere S(0,D^½) as having an index approximately 1, and points outside sphere as having index approximately 0.
> - Observe, that this behavior is not recognized by the density itself, which has no clear ‘border/change’ at the sphere S(0,D^½).
>
> We would like to thank the reviewer for pointing out that we have not explained the above reasoning in the paper clearly enough, and we shall improve it in the updated version of the paper. We would also like to add that this theoretical formula was needed only for the justification of the method, and therefore was never used in experiments.
>
> 3 & 4) The reviewer is right in noticing that most of our experiments have been performed using DCGAN, therefore the notion of “latent codes of real images” is not feasible. We did assess the method on an AE-based model; note that it was a Wasserstein Autoencoder (Figure 4), not a Variational Autoencoder, and the paths were optimized using the density-based realism index, not the FID-based. We chose WAE over VAE, as our preliminary experiments suggested it yields lower reconstruction error.
>
> On the other hand, one problem with GLOW is that it does not lead to a completely valid generative model, in the sense that the samples are not generated from the distribution which is learned during the training (the authors in GLOW use the concept of temperature, which results in sampling from the normal density with covariance smaller then identity, while they train the model using standard normal density). For this reason we did not perform any experiments with this architecture.
>
> In our preliminary experiments we have used some real data points in the VAE-type model. However, we did not observe any difference in the methods behavior in comparison to the sample-points interpolations, and therefore we did not proceed with making further experiments in this direction.
>
> 4') We use the optimization described in section 'Optimisation procedure' to reduce the number of necessary iterations in our method. This is particularly evident for a DCGAN with semicircle prior (see Fig. 1). In the initial phase of optimizing the linear interpolation, the value of the gradient for the middle points (located in the middle of the interpolation curve) can be extremely small due to the diminishing density. As a consequence, this part of the curve converges slower than the points lying closer to the ends of the curve. Our experiments for this model show that we can get the same result without using the described acceleration, but with up to 3 times more iterations. This procedure is only crucial for the semicircle prior; for the other experiments there is no significant impact on the number of iterations.
>
> We would like to thank the reviewer for drawing our attention to explaining the motivation behind this optimization more clearly; we will update the paper accordingly.

---

> > ### Author Response · Authors · 2019-11-07
> > **Response continuation**
> >
> > 5) We use the epsilon for stability reasons. The discussion on the effects of the value of the epsilon is made in section 4 (page 6): “Effects of ε-regularisation on optimisation”. In the limiting case, if epsilon is close to ½, the realism index would be equal to ½ for all points.
> >
> > We did perform the same experiments as in appendix C for the Celeb-A dataset, achieving similar results as for the MNIST dataset - therefore we included only one of them. We agree that the discussion on the effects of the epsilon regularization is an interesting topic, and we are happy to add the Celeb-A results to the appendix.

---

> > > ### Comment · AnonReviewer4 · 2019-11-14
> > > **Answers are reasonable, inclusion into the paper is needed**
> > >
> > > First and foremost, the responses do make sense to the reviewer (subject to their inclusion into the paper as the paper seems not updated yet).
> > >
> > > Re 3-4) The main point of the original statement in the review was that adding some quantitative experiments in addition to the already given qualitative ones. In the opinion of the reviewer it is the most critical point in the support of the claim of the paper.
> > >
> > > " We chose WAE over VAE, as our preliminary experiments suggested it yields lower reconstruction error." The reviewer thinks that these preliminary experiments, even in the appendix, might be instrumental in improvement of performance assessment for different types of models.

---

### Public Comment · ~Hang_Shao1 · 2019-11-01
**The authors should discuss the relationship of their work to the following paper: Shao et al., "The Riemannian Geometry of Deep Generative Models", CVPR DiffCVML Workshop 2018 (https://arxiv.org/abs/1711.08014).**

The authors should discuss the relationship of their work to the following paper: Shao et al., "The Riemannian Geometry of Deep Generative Models", CVPR DiffCVML Workshop 2018 (https://arxiv.org/abs/1711.08014). This paper also computed geodesics in the latent space of a generative model for interpolation and other purposes. Both of your strategies approximated geodesic by discretizing a curve into small intervals and optimized the problem in low dimension space, although your work is unique in introducing the realism index.

---

> ### Author Response · Authors · 2019-11-04
> **Thank you for pointing out a related paper**
>
> Thank you for your comment, and for pointing out this related work. The methods developed there are definitely related to ours, although they seem a little more complex. In the final version of our paper, we will discuss the relation between these two works in detail.

---

### Author Response · Authors · 2019-11-15
**Updates to the revised paper**

Dear Reviewers.

Thank you for your additional comments. We have updated our paper and uploaded it unto the OpenReview server. Here is a list of changes in short:

1. The FID problem (reviewer #4 / comment 1): we added a comment, see p. 8 of the paper, about the FIDs limitations and  the possible role of future better suited model. Still, the FID replacement was not the primary goal of this paper.

2. Bubble effect and the proposed realism index RI behaviour (reviewer #4 / comment 2): we added a clarification, see p. 3 of the paper, that the proposed RI index unambiguously recognises the change in density at the sphere border. That was a very good comment clarifying the RI behaviour — thank you for drawing our attention to it.

3. The auto-encoder architectures (reviewer #4 / comments 3-4): we provided an additional figure 9, see page 14, with experiments using generative auto-encoder architectures. The proposed interpolation scheme is also valid there. The presented interpolations are calculated between two encodings of real images (from the test data).

4. Optimisation of interpolation procedure (reviewer #4 / comment 4’): we added a short explanation, see footnote at page 5, why the proposed speed-up procedure of the optimisation works, especially in non-standard densities. Additionally, we added a short appendix C,  with a longer explanation and additional figure 10 for visualisation.

5. Epsilon value impact (reviewer #4 / comment 5): we added a short comment, see appendix D, about epsilon value’s impact with other datasets, where the epsilon value impact was identical.

6. First figure placement at the top of the paper  (reviewer #4 / comment 6): we decided to leave it in the same place: in our opinion a simple graph at the beginning of a paper frequently clarifies what is on the authors mind.

7. The formula in Definition 1 (reviewer #2 / comment): we added a short comment clarifying that the proposed measure is a probability density function PDF of random variable f(x).

---

### Decision · Program_Chairs · 2019-12-19

**Decision:**

Reject

**Comment:**

This paper introduces a realism metric for generated covariates and then leverage this metric to produce a novel method of interpolating between two real covariates. The reviewers found the method novel and were satisfied with the response form the authors to their concerns. However, Reviewer 4 did have reservations about the response to his/her points 3 and 4. Moreover, in the discussion period it was decided that while the method was well justified by intuition and theory, the empirical evaluation—which is the what matters at the end of the day—was unconvincing.